# Effects of total intravenous anesthesia on postoperative quality of recovery and the levels of inflammatory factors in patients undergoing retroperitoneal endoscopic surgery in urology: Study protocol for a randomized, controlled trial

**Long Qian**[1], **Hongya Liu**[1], **Caiyun Li**[1], **Yan Wang**[1], **Jianhui Li**[1], **Jingjing Pan**[1], **Wangming Li**[2]*

**1** Department of Anesthesiology, Guanyun People's Hospital, Lianyungang, Jiangsu, China, **2** Department of Anesthesiology, Lianyungang Hospital of Traditional Chinese Medicine, Lianyungang, Jiangsu, China

☯ These authors contributed equally to this work.
* lwmqxylyg@163.com

## Abstract

### Introduction

The utilisation of propofol-based total intravenous anesthesia has been demonstrated to reduce the incidence of postoperative nausea and vomiting, as well as postoperative cognitive dysfunction, in patients undergoing general anesthesia. The objective of the present study is to explore the impact of total intravenous anaesthesia on the postoperative recovery process and the levels of inflammatory factors in urology patients undergoing retroperitoneal endoscopic surgery.

### Methods

This randomized controlled clinical trial is to be carried out at the Guanyun People's Hospital, China. Eighty adult participants scheduled for urological retroperitoneoscopic surgery under general anesthesia will be randomly distributed between two groups, namely the total intravenous anesthesia group and the inhalation anesthesia group (with a total of 40 participants in each group). The random distribution of participants between the two groups will be in a 1:1 ratio. The induction of anaesthesia will be administered to all patients using a combination of propofol, cisatracurium, and sufentanil. During the maintenance of general anaesthesia, the total intravenous anesthesia group will be administered propofol and remifentanil. Conversely, the inhalation anaesthesia group will be given sevoflurane and remifentanil. The primary outcome is measured by the patient's Global Quality of Recovery-40 (QoR-40) score on post-operative day 1 (POD 1). The secondary outcomes include QoR-40 score on POD 2 and POD 3, levels of proinflammatory cytokines (tumor necrosis factor-α, interleukin-1β, and interleukin-6), pain scores at rest and while coughing, time to

**Data availability statement:** No datasets were generated or analysed during the current study. All relevant data from this study will be made available upon study completion.

**Funding:** This work will be supported by the Lianyungang Health Science and Technology Project (NO.202535). The sponsors or funders don't play any role in the study design, data collection and analysis, decision to publish, or preparation of the manuscript, No additional external funding has been received for this study.

**Competing interests:** The authors have declared that no competing interests exist.

extubate, length of patients' stay in the post-anesthesia care unit, awakening time, ramsay sedation score on awakening, adverse events (postoperative symptoms such as nausea, vomiting, headache, respiratory depression, hypoxemia). All analyses will be conducted in the modified intention-to-treat population.

## Discussion

This randomised controlled trial is designed to detect the effects of total intravenous anaesthesia on the quality of recovery after surgery and the levels of inflammatory factors in the postoperative period, as measured in patients undergoing retroperitoneoscopic urological surgery. The findings will yield insights that will contribute to effective postoperative recovery strategies in patients who undergo retroperitoneal laparoscopic surgery for urinary system-related procedures.

## Clinical Trial registration

Chinese Clinical Trial Registry (ChiCTR2500110244).

## Introduction

Compared with the transperitoneal laparoscopic approach, the retroperitoneoscopic approach has many advantages in urological surgery, such as shorter puncture path, a lower level of blood loss, reduced postoperative pain, and a faster recovery time [1–4]. However, the retroperitoneal anatomy is associated with a confined working space, which can render the procedure technically demanding. Consequently, elevated insufflation pressures (20–25 mmHg) of $CO_2$ are required to generate a more substantial working space [5,6]. More $CO_2$ is absorbed into the blood, causing harm to the human internal environment.

Propofol-based total intravenous anaesthesia (TIVA) and inhalational sevoflurane anaesthesia are two of the most common general anaesthesia techniques in clinical practice. Compared with inhalation anesthesia, whether TIVA can provide superior postoperative quality of recovery is contingent upon the nature of the surgical intervention [7–9]. The Global Quality of Recovery-40 (QoR-40) questionnaire has been developed for the specific purpose of evaluating patients' health status following surgery and anaesthesia. The QoR-40 comprises 5 domains: physical comfort, physical independence, emotional state, psychological support, and pain [10]. The QoR-40 is a reliable and valid instrument that has a broad clinical application [11,12]. Peripheral inflammatory factors can lead to inflammatory reactions such as vasodilation, increased permeability, leukocyte exudation, and even increase the permeability of the blood-brain barrier. Inflammatory factors enter the central nervous system, and neurons are damaged or die, resulting in brain tissue damage. By measuring and comparing the levels of perioperative inflammatory factors, we can predict the postoperative recovery of patients [13,14].

Accordingly, the objective of our study is to make a comparison of the effects on recovery quality of total intravenous anaesthesia and inhalation anaesthesia. These

effects will be measured using the QoR-40 questionnaire, whilst also taking into consideration the levels of inflammatory factors in patients undergoing urological retroperitoneoscopic surgery.

## Materials and methods

### Study design

The trial is of a prospective, randomised, patient- as well as assessor-blind, parallel-group controlled study design. It was approved by the Ethics Committee of the Guanyun People's Hospital (Approval No. 2025−046) on 15 September 2025. The present trial has been registered at the Chinese Clinical Trial Registry (ChiCTR2500110244; accessible at: https://www.chictr.org.cn/showproj.html?proj=288147), with the registration date of October 11, 2025. The present study will be carried out on the basis of the Declaration of Helsinki. The protocol under consideration adheres to the guidelines set out in the Standard Protocol Items: Recommendations for Interventional Trials (SPIRIT) statement (Supporting information S1 File). At the time of manuscript submission, the process of participant recruitment had not yet commenced. This trial is to be executed at the Guanyun People's Hospital in Lianyungang City, Jiangsu Province, China. A total of 80 subjects will be allocated at random to one of two groups: a group receiving total intravenous anaesthesia (n = 40) or a group receiving inhalation anaesthesia (n = 40). We plan to recruit patients from April 1st, 2026, to May 30th, 2027. All patients' written informed consent will be obtained. The study timeline and schedule of enrolment, allocation, interventions, and assessments are shown in Fig 1. The flow diagram of the study is shown in Fig 2.

An independent researcher will conduct the randomisation, and the random number will be generated by the computer after the assessment of eligibility. The details of randomisation are to be stored in an opaque sealed envelope. Two hours before the operation, the patient's anesthesiologist and nurse will open the opaque sealed envelope in turn and treat the patient according to the allocation. Because of the significant difference in anesthetic drugs, the general anaesthesia will be performed by the only unblinded staff, the anesthesiologists. However, they will not be involved in postoperative care or the assessment of postoperative outcomes. Throughout the study period, neither the participants nor the surgeons will be aware of the group allocation. The medical personnel who administered post-operative treatment and assessed the outcomes in the post-anaesthesia care unit (PACU) and ward are also oblivious to the group allocation.

### Eligibility criteria

**Inclusion criteria.** Patients aged 18–65 with a physical status I-III according to the American Society of Anesthesiologists (ASA), who are scheduled to undergo urological retroperitoneoscopic surgery under general anaesthesia will be considered for inclusion.

**Exclusion criteria.** The exclusion of patients from the study will be based on their meeting one or more of the following exclusion criteria: (1) unplanned or emergency surgery; (2) those with mental illness, cognitive dysfunction, chronic pain disease, long-term use of sedative and analgesic drugs or alcohol abuse; (3) severe liver and kidney disease;(4) severe cardiopulmonary disease; (5) pregnant women and lactating women; or (6) those who are allergic or contraindicated to the drugs used in the research.

**Withdrawal criteria.** Withdrawal criteria include: (1) withdrawing consent or request participation termination during the observation period; (2) subjects or researchers know the experimental grouping rate is more than 20%; or (3) $PaCO_2 \geq 90$ mmHg and $PH \leq 7.15$ during the operation.

### Study implementation

**Anesthesia and monitoring methods.** All patients will be fasting for 6–8 h and water deprivation for 2 h before the operation, and will not receive any preoperative medication. Baseline blood pressure and heart rate will be obtained in the preoperative preparation area. Following their entry into the operating theatre, patients will be monitored in accordance

| Time point | Enrolment — Preoperative visit | Allocation — 2 hours before surgery | Post-allocation — intraoperatively | PACU | POS 2h | POS 6h | POS 24h | POS 48h | POS 72h |
|---|---|---|---|---|---|---|---|---|---|
| **Patient enrolment** | | | | | | | | | |
| Eligibility criteria | × | | | | | | | | |
| Written informed consent | × | | | | | | | | |
| Demographic data | × | | | | | | | | |
| Baseline characteristics | × | | | | | | | | |
| Randomisation/allocation | | × | | | | | | | |
| **Study interventions** | | | | | | | | | |
| Total intravenous anaesthesia | | | × | | | | | | |
| Inhalation anesthesia | | | × | | | | | | |
| **Outcome assessment** | | | | | | | | | |
| QoR-40 scores | × | | | | | | × | × | × |
| Postoperative pain scores | | | | | × | × | × | | |
| Length of PACU stay | | | | × | | | | | |
| Levels of proinflammatory cytokines | × | | | | × | × | × | | |
| Hemodynamic parameters | | | × | | | | | | |
| ETCO2 | | | × | | | | | | |
| PaCO2 | | | × | | | | | | |
| Awakening time | | | × | | | | | | |
| Extubation time | | | × | | | | | | |
| Level of sedation | | | × | | | | | | |
| PONV | | | | | | | × | | |
| Headache | | | | | | | × | | |
| Hypoxemia | | | | | | | × | | |
| Respiratory depression | | | | | | | × | | |

**Fig 1. Patient enrolment schedule.** QoR-40, 40 items of Quality of Recovery; PACU, post-anaesthesia care unit; PONV, postoperative nausea and vomiting; POS, postoperative; ETCO2, end-tidal carbon dioxide; PaCO2, partial pressure of arterial carbon dioxide.

with standard protocol, which includes electrocardiography (ECG) and pulse oximetry (SpO2). The radial artery will be cannulated for the purpose of continuous measurement of arterial pressure. The depth of anaesthesia will be measured using the Bispectral Index (BIS). Monitoring of end-tidal carbon dioxide concentration (ETCO2) will be conducted post-intubation. Participants will be divided into two groups, namely group T and group C, according to the allocation procedure. Group T will receive total intravenous anaesthesia, while group C will receive inhalation anaesthesia.

**Intravenous anesthesia induction.** It is imperative that all patients receive a continuous intravenous infusion of 0.9% sodium chloride at a rate of 6–8 mL/kg/h. Furthermore, patients must undergo preoxygenation with 100%

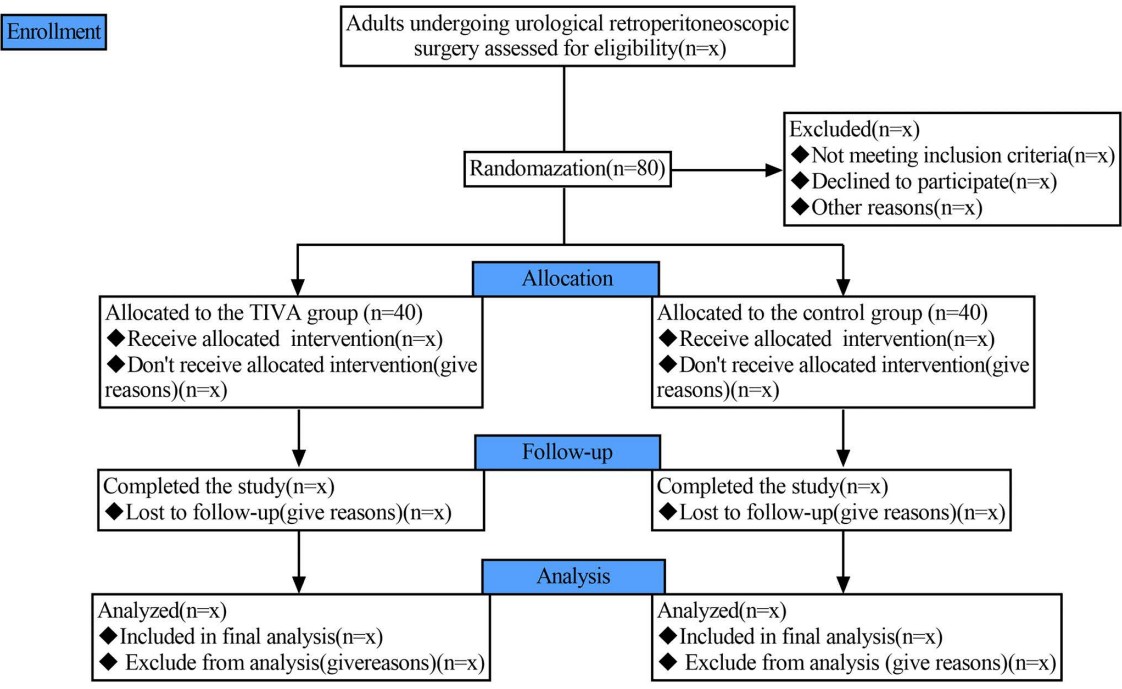

**Fig 2. Study flow diagram.** TIVA, total intravenous anesthesia.

oxygen at a rate of 6 L/min via a mask during the induction period. The two groups will receive the same anesthesia induction methods: 1.5–2 mg/kg of propofol. In the event of the Bis value being ≤ 60, the recommended administration is 2 mg/kg of cisatracurium and 0.5 µg/kg of sufentanil. In the event of the BIS value being higher than 60, a bolus of 1 mg/kg of propofol should be administered immediately, with a minimum interval of one minute between each subsequent bolus administration, until the BIS value reaches a level lower than 60. Volume-controlled ventilation will be performed (tidal volume 6–8 ml/kg, respiratory rate 12–18 times/min, and PEEP 5 cmH$_2$O) after endotracheal intubation.

**Anesthesia maintenance.** Anesthesia maintenance is defined as the time period commencing from the completion of induction and concluding at the moment of the cessation of anaesthesia (the immediate cessation of the infusion or inhalation of anaesthetic drugs following the conclusion of surgical procedures). The infusion of propofol at a rate of 4–12 mg/kg/h for maintenance purposes will be administered to Group T, whereas Group C will undergo inhalation of sevoflurane at 1.0-3.0% for maintenance purposes. The intravenous infusion of remifentanil will be initiated at a rate of 0.05–0.2 µg/kg/min, with cisatracurium and sufentanil being added according to the requirements of the surgical procedure in both groups. The depth of anaesthesia will be adjusted until the Bispectral Index (BIS) values are within the desired range of 40–55, by means of manual adjustments to the infusion of propofol or the inhalation of sevoflurane. Upon completion of the surgery, all anesthetic agents will be discontinued. The provision of patient-controlled intravenous analgesia (PCIA) is to be administered to all patients, with the dosing regimen being 1.5 µg/kg of sufentanil and 0.9% normal saline, with a total volume of 100 ml, at a rate of 2 ml/h for a duration of 48 hours. It is asserted that the self-controlled capacity will be 0.5 ml and that the locking time will be 15 minutes.

**Recovery of anesthesia.** Subsequent to the extubation process, patients are to be transferred to a post-anaesthesia care unit (PACU), where they will be provided with oxygen supplementation of 3 L/min via a nasal catheter. The attainment of a Steward awakening score of 4 points or more will result in the patient's discharge from the Post-Anaesthetic Care Unit to the surgical wards.

## Outcomes

**Primary outcomes.** The primary outcome of this study is the Global Quality of Recovery-40 (QoR-40) score on Post-Operative Day 1 (POD 1). The QoR-40 is a scale with 40 items that assesses five aspects of recovery: emotions, physical comfort, psychological support, physical independence, and pain. Scores on the scale range from 40 to 200. Clinicians and researchers have widely used QoR-40 in various surgical contexts and anaesthetic techniques.

**Secondary outcomes.** The secondary outcomes encompass the QoR-40 score on POD 2 and POD 3, in addition to the levels of pro-inflammatory cytokines (tumour necrosis factor-α, interleukin-1β, and interleukin-6), hemodynamic parameters, arterial and end-tidal carbon dioxide pressure, pain scores at rest and while coughing, extubation time, length of patients' stay in the post-anesthesia care unit, Ramsay sedation score on awakening, awakening time, and adverse events (postoperative symptoms such as nausea, vomiting, headache, respiratory depression, and hypoxemia).

## Sample size estimation

The primary endpoint that will be evaluated in this study is the QoR-40 score on the first postoperative day. Previous studies reported that the minimal clinically important difference (MCID) of the QoR-40 score was 6.3 points [10]. Based on our institutional data, the data were normally distributed and exhibited homogeneity in variance. The standard deviation was 9.3 points, and the sample size was calculated using the two-sample t-test. On the basis of this assumption, it is estimated that 36 patients in each group would be required, with a power of 80% and at an α level of 0.05. To allow for a 10% possible dropout, a total of 80 patients (n = 40 in each group) will be enrolled. The sample size is calculated using the PASS software.

## Data collection and registration

The collection of patients' demographic data and baseline characteristics is to be carried out by independent investigators who have undergone the relevant training and who are unaware of group assignment. The collection of primary and secondary outcome measures, in addition to other non-outcome data pertaining to the perioperative period, is to be undertaken. The values of the following parameters will be recorded at five distinct time points for the duration of the study: MAP, HR, BIS, ETCO2, and PaCO2. The parameters will be recorded at the following time points: baseline, 30 minutes into the surgical procedure, 60 minutes into the surgical procedure, the conclusion of the surgical procedure, and immediately post-tracheal extubation. The evaluation of postoperative pain will be conducted utilising the Numeric Rating Scale (NRS), which ranges from 0 to 10 (0 representing no pain, while 10 denotes the most severe pain conceivable). Levels of proinflammatory cytokines and NRS will be recorded at postoperative 2, 6, 24 h. Data collection will be performed using electronic case report forms (eCRFs) established on the Clinical Trial Management Public Platform. Standard data management procedures will be used to manage all data. This includes double data entry, data validation, query management, and database locking. These procedures will be followed before statistical analysis. To protect participant privacy, de-identification will be conducted by removing all personally identifiable information. Comprehensive data security measures and access control policies will be implemented throughout the study. These include role-based access authorization, encrypted data transmission and storage, regular data backup, and complete audit trails. These measures will ensure data confidentiality, integrity, and traceability. It is the responsibility of the investigators to conduct an interim analysis to make an informed decision regarding the continuation of the trial or the modification of the study design with a view to optimising the trial results.

## Statistical analysis

The demographic data and baseline characteristics will only be expressed using descriptive statistics and will not be compared between groups. According to standard statistical reasoning, testing for demographic and baseline balance is

unnecessary and uninformative in randomized controlled trials, because it only evaluates random variation, not clinically significant imbalance. Continuous data will be checked for the normality of the distribution with the Shapiro-Wilk test. Data conforming to the normal distribution will be expressed as means with standard deviations and analyzed using an independent samples t-test. Data with the non-normal distribution will be reported as medians with interquartile ranges and examined using the Mann-Whitney rank sum test. Categorical data will be presented as counts, along with the corresponding percentages, and between-group differences will be assessed using the Chi-square test or Fisher's exact test when appropriate. For normally distributed continuous variables measured at multiple time points, a repeated measures analysis of variance will be performed. This will be accompanied by Mauchly's test of sphericity. Greenhouse-Geisser correction will be used when the sphericity assumption is violated. For repeated measures of continuous data that failed to meet normality assumptions, linear mixed-effects models will be constructed to estimate the effects of group, time, and the group-by-time interaction, with random intercepts adjusted for each subject to account for intra-individual correlation. All outcome measures will be analyzed using an intention-to-treat population, encompassing all patients randomized to the respective groups. Missing data will be imputed using the appropriate method, such as averaging the values within the group or using multiple imputation. Multiple comparison corrections for the secondary outcomes will not be performed, so these outcomes should be regarded as exploratory. The independent statistician will use SPSS software to analyse the data. The existence of a statistically significant difference is indicated by a two-sided P value $< 0.05$.

## Ethical considerations and declarations

The present study will be conducted in strict accordance with the Declaration of Helsinki and has been approved by the Ethics Committee of the Guanyun People's Hospital (approval number: 2025-046). Prior to participating in the study, written informed consent will be obtained from all subjects and/or their legal representatives. It is the responsibility of the Investigator to report the occurrence of all Serious Adverse Events (SAEs) to Wangming Li (the corresponding author in the trial) without delay, and to apprise the Ethics Committee accordingly. The organiser is tasked with the continuous evaluation of the anaesthesia protocol's safety. In the event of an adverse event (AE), the test will be terminated, and the participants will receive compensation. The results of this study are scheduled for publication in a peer-reviewed scientific periodical.

**Informed consent.** Prior to undergoing surgery, each participant will be requested to provide written consent, a process that will be overseen by the research team. The consent process will encompass a comprehensive explanation of the clinical trial's purpose, methodologies, anticipated benefits, and potential risks. Should the findings of the new safety study indicate a significant alteration to the benefit-risk assessment, the informed consent form will be subjected to a thorough review and modification. Furthermore, all study participants will be furnished with access to the revised information. Henceforth, participants will be required to provide their signed consent in the form of the revised informed consent form. Researchers are required to obtain consent in order to ensure the continuity of their research project. Study participants will be subjected to screening in accordance with the established inclusion and exclusion criteria. This process will take place following the signing of the informed consent form.

## Discussion

This present randomised controlled trial will recruit 80 adult patients undergoing urological retroperitoneoscopic surgery, with the aim of evaluating the effects of total intravenous anaesthesia versus inhalation anaesthesia on postoperative recovery quality and inflammatory factors. Furthermore, a comparative analysis will be conducted to explore the divergent outcomes elicited by the two anaesthetic techniques, with a focus on pain scores, extubation time, length of patients' stay in the post-anesthesia care unit, and postoperative sedation. This study will be conducted on the basis of the Consolidated Standards of Reporting Trials guidelines [15].

As demonstrated in the extant literature, total intravenous anaesthesia has been utilised in a number of surgical contexts to assess the effects of postoperative recovery quality. Lee WK and colleagues reported that in female thyroid surgery, the QoR-40 score on POD1 was significantly higher in the total intravenous anesthesia group compared with the inhalation anesthesia group, especially the dimensions of physical comfort and physical independence [8]. Elbakry AE and colleagues suggested that the use of total intravenous anaesthesia may facilitate enhanced postoperative recovery, reduced postoperative adverse effects, and diminished analgesic requirements in morbidly obese patients undergoing laparoscopic sleeve gastrectomy [16]. Liu T and colleagues reported that total intravenous anaesthesia could improve the QoR-40 score at 6 hours after operation in the patients undergoing endoscopic sinus surgery [17]. Nevertheless, Niu Z et al. demonstrated that their total intravenous anaesthesia did not impact the overall postoperative recovery of patients who underwent total laparoscopic hysterectomy, as evidenced by the utilisation of the QoR-40 questionnaire [9]. Consequently, the effects of the administration of total intravenous anaesthesia are found to be inconsistent according to differing surgical types.

Streich B and colleagues showed that compared with intraperitoneal carbon dioxide insufflation, retroperitoneal insufflation causes more carbon dioxide absorption, and hypercapnia should be avoided by increasing the controlled ventilation [18]. Fraser S and colleagues reported that in patients undergoing posterior retroperitoneoscopic adrenalectomy, higher insufflation pressure was associated with higher arterial and end tidal carbon dioxide partial pressures, as well as lower pH at 60 minutes [19].In this study, we will measure the arterial and end tidal carbon dioxide partial pressures at different times during the operation to investigate whether total intravenous anaesthesia can reduce the carbon dioxide absorption. Considering the anti-inflammatory properties of propofol, total intravenous anaesthesia based on propofol may be used as a feasible scheme to reduce the risk of postoperative cognitive dysfunction [20]. Yıldız MB et al. demonstrated that, in patients subjected to lumbar microdiscectomy, postoperative interleukin-6 (IL-6) levels were markedly diminished in the total intravenous anaesthesia group. This finding suggests that total intravenous anaesthesia with propofol may provide a more effective modulation of inflammatory responses [21]. However, Sofra M. and colleagues' findings revealed that the immunomodulatory effects of intravenous and balanced inhalation anaesthetics in patients undergoing elective radical cystectomy were comparable and did not differ significantly [22]. Hence, in this study, we will measure the levels of inflammatory factors during the perioperative period to explore whether total intravenous anesthesia can reduce the perioperative inflammatory response.

This study has some limitations. First, the calculation of the required sample size is predicated upon the QoR-40 score on POD 1. It is imperative to interpret the outcomes of inflammatory factors and other secondary outcomes as exploratory in nature. Second, it is impossible for the anesthesiologist to be unaware of the fact that there are differences between techniques. Nevertheless, their involvement is limited to this particular component of the study, and they will not contribute to any other aspects of the research. Third, the primary outcome of the QoR-40 score is reported by patients themselves. Finally, given that this is a single-centre trial, further studies are required to evaluate the impact of total intravenous anaesthesia on the quality of postoperative recovery and to confirm the findings of this study.

In summary, the present randomised controlled trial will detect the effects of total intravenous anaesthesia on the quality of recovery after surgery and on inflammatory factors following retroperitoneoscopic urological surgery. The findings of the present study have the potential to engender a novel approach to enhance the postoperative recovery of patients who have undergone retroperitoneal laparoscopic surgery for urinary system-related conditions.

## Supporting information

**S1 File. SPIRIT checklist.**
(DOC)

**S2 File. Study protocol in English.**
(DOCX)

**S3 File. Ethics approved protocol Chinese.**
(DOC)

## Acknowledgements

The authors would like to express their gratitude to the patients who will participate in this study, as well as to the medical, nursing, and research teams.

## Author contributions

**Conceptualization:** Long Qian, Hongya Liu, Wangming Li.

**Data curation:** Long Qian, Hongya Liu, Caiyun Li, Yan Wang, Jianhui Li.

**Formal analysis:** Jingjing Pan.

**Funding acquisition:** Long Qian, Wangming Li.

**Investigation:** Long Qian, Caiyun Li, Yan Wang, Jianhui Li.

**Methodology:** Long Qian, Hongya Liu, Caiyun Li, Yan Wang, Jianhui Li.

**Project administration:** Long Qian, Hongya Liu, Caiyun Li, Yan Wang, Jianhui Li, Wangming Li.

**Supervision:** Wangming Li.

**Writing – original draft:** Long Qian, Hongya Liu, Wangming Li.

**Writing – review & editing:** Long Qian, Hongya Liu, Wangming Li.

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
