## [Decision Letter · Decision Letter 0]

8 Feb 2026

PONE-D-25-62547Effects of total intravenous anesthesia on postoperative quality of recovery and the levels of inflammatory factors in patients undergoing retroperitoneal endoscopic surgery in urology: Study protocol for a randomized, controlled trialPLOS One

Dear Dr. li,

Thank you for submitting your manuscript to PLOS ONE. After careful consideration, we feel that it has merit but does not fully meet PLOS ONE’s publication criteria as it currently stands. Therefore, we invite you to submit a revised version of the manuscript that addresses the points raised during the review process.

The manuscript has been evaluated by two reviewers, and their comments are available below.

The reviewers have raised a number of concerns. They feel the manuscript should clearly state the determination of sample size, and more clearly articulate the planned statistical analysis once data collection is complete. They also request some improvements to the language and reporting before we can accept your study protocol for publication.

Could you please carefully revise the manuscript to address all comments raised?

If applicable, we recommend that you deposit your laboratory protocols in protocols.io to enhance the reproducibility of your results. Protocols.io assigns your protocol its own identifier (DOI) so that it can be cited independently in the future. For instructions see: https://journals.plos.org/plosone/s/submission-guidelines#loc-laboratory-protocols. Additionally, PLOS ONE offers an option for publishing peer-reviewed Lab Protocol articles, which describe protocols hosted on protocols.io. Read more information on sharing protocols at . Additionally, PLOS ONE offers an option for publishing peer-reviewed Lab Protocol articles, which describe protocols hosted on protocols.io. Read more information on sharing protocols at https://plos.org/protocols?utm_medium=editorial-email&utm_source=authorletters&utm_campaign=protocols..

We look forward to receiving your revised manuscript.

Kind regards,

Jennifer Tucker, PhD

Staff Editor

PLOS One

Journal Requirements:

[This work will be supported by the Lianyungang Health Science and Technology Project（NO.202535）.The sponsors or funders don't play any role in the study design, data collection and analysis, decision to publish, or preparation of the manuscript,].

Please provide an amended statement that declares *all* the funding or sources of support (whether external or internal to your organization) received during this study, as detailed online in our guide for authors at http ://journals.plos.org/plosone/s/submit-now. Please also include the statement “There was no additional external funding received for this study.” in your updated Funding Statement.

[This work will be supported by the Lianyungang Health Science and Technology Project (NO.202535). The funding agency will have no input in the design of the research, nor will they participate in preparing or submitting manuscripts for publication.]

[This work will be supported by the Lianyungang Health Science and Technology Project（NO.202535）.The sponsors or funders don't play any role in the study design, data collection and analysis, decision to publish, or preparation of the manuscript,]

Reviewers' comments:

Reviewer's Responses to Questions

**Comments to the Author**

1. Does the manuscript provide a valid rationale for the proposed study, with clearly identified and justified research questions?

Reviewer #1: Partly

Reviewer #2: Yes

2. Is the protocol technically sound and planned in a manner that will lead to a meaningful outcome and allow testing the stated hypotheses?

Reviewer #1: No

Reviewer #2: Yes

3. Is the methodology feasible and described in sufficient detail to allow the work to be replicable?

Reviewer #1: No

Reviewer #2: Yes

4. Have the authors described where all data underlying the findings will be made available when the study is complete?

The PLOS Data policy requires authors to make all data underlying the findings described in their manuscript fully available without restriction, with rare exception, at the time of publication. The data should be provided as part of the manuscript or its supporting information, or deposited to a public repository. For example, in addition to summary statistics, the data points behind means, medians and variance measures should be available. If there are restrictions on publicly sharing data—e.g. participant privacy or use of data from a third party—those must be specified.requires authors to make all data underlying the findings described in their manuscript fully available without restriction, with rare exception, at the time of publication. The data should be provided as part of the manuscript or its supporting information, or deposited to a public repository. For example, in addition to summary statistics, the data points behind means, medians and variance measures should be available. If there are restrictions on publicly sharing data—e.g. participant privacy or use of data from a third party—those must be specified.

Reviewer #1: Yes

Reviewer #2: No

5. Is the manuscript presented in an intelligible fashion and written in standard English?

Reviewer #1: No

Reviewer #2: Yes

6. Review Comments to the Author

You may also provide optional suggestions and comments to authors that they might find helpful in planning their study.

Reviewer #1: This is a well-designed and ethically approved randomized controlled trial protocol addressing a relevant clinical question. The primary outcome selection and sample size justification are appropriate. Prior to acceptance, the authors should clarify the hierarchy of outcomes, strengthen the pre-specified statistical analysis plan, and address minor language issues. These revisions would improve transparency and methodological rigor in line with PLOS ONE requirements.

Reviewer #2: This protocol describes a single-center, randomized, patient- and assessor-blinded controlled trial investigating whether total intravenous anesthesia (TIVA) improves postoperative recovery compared with inhalation anesthesia in patients undergoing urological retroperitoneoscopic surgery. A total of 80 adult patients (ASA I–III) will be randomized 1:1 to receive either propofol–remifentanil–based TIVA or sevoflurane–remifentanil inhalation anesthesia. The primary outcome is postoperative recovery quality measured by the QoR-40 score on postoperative day 1. Secondary outcomes include QoR-40 scores on days 2 and 3, perioperative inflammatory markers (TNF-α, IL‑1β, IL‑6), pain scores, extubation and awakening times, PACU length of stay, sedation levels, and adverse events. The study aims to clarify whether TIVA enhances recovery quality and attenuates inflammatory responses following retroperitoneoscopic urological surgery.

Minor Suggested Revisions:

1- Sample Size Justification: Explicitly state the statistical test underlying the sample size calculation (e.g., two-sample t-test) and clarify any assumptions regarding normality and equal variances between groups.

2- Handling of Secondary Outcomes and Multiplicity: Multiple secondary outcomes are planned, including repeated QoR‑40 measurements, inflammatory biomarkers, pain scores, and several perioperative endpoints. The protocol does not address whether adjustments for multiple comparisons will be applied or whether secondary outcomes are considered exploratory. Clarification is recommended to avoid inflation of type I error and overinterpretation of secondary findings.

3- Repeated Measures Analysis: QoR‑40 scores and inflammatory markers are measured at multiple postoperative time points. The authors propose repeated-measures ANOVA but do not specify how model assumptions (e.g., sphericity) will be assessed or whether alternative methods (e.g., mixed-effects models) would be considered if assumptions are violated.

4- Data Management and Storage: Data collection using electronic case report forms (eCRFs), database locking, and de-identification are described. However, the protocol does not specify the data management platform or system used for eCRFs, nor does it describe data security or access controls. Providing this information would strengthen data governance transparency.

5- Baseline Characteristics: The protocol states that baseline characteristics will be presented descriptively but not statistically compared between groups. This approach is acceptable and consistent with CONSORT recommendations; however, a brief justification for this decision would be helpful for clarity.

6- Data Summary: Clarify that categorial data will be presented as counts with corresponding percentages.

7. PLOS authors have the option to publish the peer review history of their article (what does this mean?). If published, this will include your full peer review and any attached files.). If published, this will include your full peer review and any attached files.

.

Reviewer #1: No

Reviewer #2: No

To ensure your figures meet our technical requirements, please review our figure guidelines: http s://journals.plos.org/plosone/s/figures

You may also use PLOS’s free figure tool, NAAS, to help you prepare publication quality figures: http s://journals.plos.org/plosone/s/figures#loc-tools-for-figure-preparation.

---

## [Author Response · Author response to Decision Letter 1]

18 Feb 2026

Response to Reviewers

We greatly appreciate the reviewers for their thoughtful comments and constructive suggestions, which have significantly improved the quality of our manuscript. We have carefully addressed all comments and revised the manuscript accordingly.

Reviewer #1: This is a well-designed and ethically approved randomized controlled trial protocol addressing a relevant clinical question. The primary outcome selection and sample size justification are appropriate. Prior to acceptance, the authors should clarify the hierarchy of outcomes, strengthen the pre-specified statistical analysis plan, and address minor language issues.

Response: We greatly appreciate the reviewer’s positive comments and constructive suggestions on our randomized controlled trial protocol. Following the reviewer’s advice, we have clarified the hierarchy of outcomes in the revised manuscript, strengthened the pre-specified statistical analysis plan, and carefully revised the manuscript to address minor language issues throughout the text. We have clarified these in the revision, “Continuous data will be checked for the normality of the distribution with the Shapiro-Wilk test. Data conforming to the normal distribution will be expressed as means with standard deviations and analyzed using an independent samples t-test. Data with the non-normal distribution will be reported as medians with interquartile ranges and examined using the Mann-Whitney rank sum test. Categorical data will be presented as counts, along with the corresponding percentages, and between-group differences will be assessed using the Chi-square test or Fisher's exact test when appropriate. For normally distributed continuous variables measured at multiple time points, a repeated measures analysis of variance will be performed. This will be accompanied by Mauchly's test of sphericity. Greenhouse-Geisser correction will be used when the sphericity assumption is violated. For repeated measures of continuous data that failed to meet normality assumptions, linear mixed-effects models will be constructed to estimate the effects of group, time, and the group-by-time interaction, with random intercepts adjusted for each subject to account for intra-individual correlation. All outcome measures will be analyzed using an intention-to-treat population, encompassing all patients randomized to the respective groups. Missing data will be imputed using the appropriate method, such as averaging the values within the group or using multiple imputation. Multiple comparison corrections for the secondary outcomes will not be performed, so these outcomes should be regarded as exploratory.” (See page 14 lines 284-296, page 15 lines 29-396). We hope these revisions meet the reviewer’s expectations and improve the quality of the manuscript.

Reviewer #2: This protocol describes a single-center, randomized, patient- and assessor-blinded controlled trial investigating whether total intravenous anesthesia (TIVA) improves postoperative recovery compared with inhalation anesthesia in patients undergoing urological retroperitoneoscopic surgery. A total of 80 adult patients (ASA I–III) will be randomized 1:1 to receive either propofol–remifentanil–based TIVA or sevoflurane–remifentanil inhalation anesthesia. The primary outcome is postoperative recovery quality measured by the QoR-40 score on postoperative day 1. Secondary outcomes include QoR-40 scores on days 2 and 3, perioperative inflammatory markers (TNF-α, IL 1β, IL 6), pain scores, extubation and awakening times, PACU length of stay, sedation levels, and adverse events. The study aims to clarify whether TIVA enhances recovery quality and attenuates inflammatory responses following retroperitoneoscopic urological surgery.

We sincerely appreciate the reviewer for the careful and constructive comments on our study protocol. We have carefully addressed each concern as detailed below.

1. Sample Size Justification: Explicitly state the statistical test underlying the sample size calculation (e.g., two-sample t-test) and clarify any assumptions regarding normality and equal variances between groups.

Response: We agree with the reviewer’s valuable comment. In the revised manuscript, we have explicitly stated that the sample size calculation was based on a two-sample t-test for the primary outcome (QoR-40 score on postoperative day 1). We have also clarified the assumptions of normality and homogeneity of variances between groups that were applied in the sample size estimation process. The relevant description has been added to the sample size section, “The primary endpoint that will be evaluated in this study is the QoR-40 score on the first postoperative day. Previous studies reported that the minimal clinically important difference (MCID) of the QoR-40 score was 6.3 points[10]. Based on our institutional data, the data were normally distributed and exhibited homogeneity in variance. The standard deviation was 9.3 points, and the sample size was calculated using the two-sample t-test. On the basis of this assumption, it is estimated that 36 patients in each group would be required, with a power of 80% and at an α level of 0.05. To allow for a 10% possible dropout, a total of 80 patients (n = 40 in each group) will be enrolled.” (See page 12 lines 239-245)

2. Handling of Secondary Outcomes and Multiplicity: Multiple secondary outcomes are planned, including repeated QoR 40 measurements, inflammatory biomarkers, pain scores, and several perioperative endpoints. The protocol does not address whether adjustments for multiple comparisons will be applied or whether secondary outcomes are considered exploratory. Clarification is recommended to avoid inflation of type I error and overinterpretation of secondary findings.

Response: We thank the reviewer for highlighting this important methodological issue. We have now clearly stated in the revised protocol that all secondary outcomes are considered exploratory, and no formal adjustment for multiple comparisons will be performed to avoid overcorrection in this exploratory analysis, “Multiple comparison corrections for the secondary outcomes will not be performed, so these outcomes should be regarded as exploratory.” (See page 15 lines 305-306)

3. Repeated Measures Analysis: QoR‑40 scores and inflammatory markers are measured at multiple postoperative time points. The authors propose repeated-measures ANOVA but do not specify how model assumptions (e.g., sphericity) will be assessed or whether alternative methods (e.g., mixed-effects models) would be considered if assumptions are violated.

Response: We appreciate the reviewer’s professional suggestion. In the revised statistical analysis plan, we have specified that sphericity will be tested using Mauchly’s test for repeated-measures ANOVA. If the sphericity assumption is violated, Greenhouse–Geisser correction will be applied. In addition, we have noted that linear mixed-effects models will be considered as an alternative method for repeated measures data of continuous data that failed to meet normality assumptions, “For normally distributed continuous variables measured at multiple time points, a repeated measures analysis of variance will be performed. This will be accompanied by Mauchly's test of sphericity. Greenhouse-Geisser correction will be used when the sphericity assumption is violated. For repeated measures of continuous data that failed to meet normality assumptions, linear mixed-effects models will be constructed to estimate the effects of group, time, and the group-by-time interaction, with random intercepts adjusted for each subject to account for intra-individual correlation.” (See page 14 lines 292-296, page 15 lines 297-301)

4. Data Management and Storage: Data collection using electronic case report forms (eCRFs), database locking, and de-identification are described. However, the protocol does not specify the data management platform or system used for eCRFs, nor does it describe data security or access controls. Providing this information would strengthen data governance transparency.

Response: We thank the reviewer for this constructive comment. In the revised protocol, we have clearly specified that data collection will be performed using electronic case report forms (eCRFs) established on the Clinical Trial Management Public Platform. We have also supplemented detailed descriptions of data security measures, access control policies, user authorization levels, and database management procedures to enhance transparency and rigor in data governance,“Standard data management procedures will be used to manage all data. This includes double data entry, data validation, query management, and database locking. These procedures will be followed before statistical analysis. To protect participant privacy, de-identification will be conducted by removing all personally identifiable information. Comprehensive data security measures and access control policies will be implemented throughout the study. These include role-based access authorization, encrypted data transmission and storage, regular data backup, and complete audit trails. These measures will ensure data confidentiality, integrity, and traceability.” (See page 13 lines 264-173)

5. Baseline Characteristics: The protocol states that baseline characteristics will be presented descriptively but not statistically compared between groups. This approach is acceptable and consistent with CONSORT recommendations; however, a brief justification for this decision would be helpful for clarity.

Response: We appreciate the reviewer’s suggestion. As noted by the reviewer, not performing statistical comparisons of baseline characteristics is consistent with CONSORT guidelines for randomized controlled trials, because randomization is expected to balance baseline variables, and significance testing of baseline imbalance is generally discouraged. We have added this brief justification in the revised manuscript to clarify the rationale for descriptive presentation only, “According to standard statistical reasoning, testing for demographic and baseline balance is unnecessary and uninformative in randomized controlled trials, because it only evaluates random variation, not clinically significant imbalance.” (See page 14 lines 281-284)

6. Data Summary: Clarify that categorial data will be presented as counts with corresponding percentages.

Response: Thanks for the reviewer’s suggestion. We have revised the data summary section to explicitly state that categorical variables will be summarized as counts, along with the corresponding percentages, “Categorical data will be presented as counts, along with the corresponding percentages, and between-group differences will be assessed using the Chi-square test or Fisher's exact test when appropriate.” (See page 14 lines 289-292)

---

## [Decision Letter · Decision Letter 1]

9 Mar 2026

PONE-D-25-62547R1Effects of total intravenous anesthesia on postoperative quality of recovery and the levels of inflammatory factors in patients undergoing retroperitoneal endoscopic surgery in urology: Study protocol for a randomized, controlled trialPLOS One

Dear Dr. li,

Thank you for submitting your manuscript to PLOS ONE. After careful consideration, we feel that it has merit but does not fully meet PLOS ONE’s publication criteria as it currently stands. Therefore, we invite you to submit a revised version of the manuscript that addresses the points raised during the review process.

The reviewers are positive about the manuscript following the revisions made. Before we proceed, could you please: i) upload the original Chinese-language IRB-approved study protocol? This is in addition to the translated protocol uploaded as Supplemental File S2. ii) On line 419: please correct a typographical error in this line and ensure that 'parent' is changed to 'patient'. Thank you for addressing these points.

If applicable, we recommend that you deposit your laboratory protocols in protocols.io to enhance the reproducibility of your results. Protocols.io assigns your protocol its own identifier (DOI) so that it can be cited independently in the future. For instructions see: https://journals.plos.org/plosone/s/submission-guidelines#loc-laboratory-protocols. Additionally, PLOS ONE offers an option for publishing peer-reviewed Lab Protocol articles, which describe protocols hosted on protocols.io. Read more information on sharing protocols at . Additionally, PLOS ONE offers an option for publishing peer-reviewed Lab Protocol articles, which describe protocols hosted on protocols.io. Read more information on sharing protocols at https://plos.org/protocols?utm_medium=editorial-email&utm_source=authorletters&utm_campaign=protocols..

We look forward to receiving your revised manuscript.

Kind regards,

Alejandro Torrado Pacheco, PhD

Associate Editor

PLOS One

Journal Requirements:

Reviewers' comments:

Reviewer's Responses to Questions

**Comments to the Author**

1. Does the manuscript provide a valid rationale for the proposed study, with clearly identified and justified research questions?

Reviewer #1: Yes

Reviewer #2: Yes

2. Is the protocol technically sound and planned in a manner that will lead to a meaningful outcome and allow testing the stated hypotheses?

Reviewer #1: Yes

Reviewer #2: Yes

3. Is the methodology feasible and described in sufficient detail to allow the work to be replicable?

Reviewer #1: Yes

Reviewer #2: Yes

4. Have the authors described where all data underlying the findings will be made available when the study is complete?

The PLOS Data policy requires authors to make all data underlying the findings described in their manuscript fully available without restriction, with rare exception, at the time of publication. The data should be provided as part of the manuscript or its supporting information, or deposited to a public repository. For example, in addition to summary statistics, the data points behind means, medians and variance measures should be available. If there are restrictions on publicly sharing data—e.g. participant privacy or use of data from a third party—those must be specified.requires authors to make all data underlying the findings described in their manuscript fully available without restriction, with rare exception, at the time of publication. The data should be provided as part of the manuscript or its supporting information, or deposited to a public repository. For example, in addition to summary statistics, the data points behind means, medians and variance measures should be available. If there are restrictions on publicly sharing data—e.g. participant privacy or use of data from a third party—those must be specified.

Reviewer #1: Yes

Reviewer #2: No

5. Is the manuscript presented in an intelligible fashion and written in standard English?

Reviewer #1: Yes

Reviewer #2: Yes

6. Review Comments to the Author

You may also provide optional suggestions and comments to authors that they might find helpful in planning their study.

Reviewer #1: Thank you for the revisions made to the manuscript. The revised study protocol is clearer and methodologically stronger compared with the previous version. In particular, the clarification of outcome hierarchy, expansion of the statistical analysis plan, and improved description of data management procedures have significantly enhanced the quality and transparency of the protocol.

However, I would suggest you change the date of the patient recruitment, which you mentioned was December 2025.

Reviewer #2: All comments have been thoroughly and adequately addressed.

7. PLOS authors have the option to publish the peer review history of their article (what does this mean?). If published, this will include your full peer review and any attached files.). If published, this will include your full peer review and any attached files.

.

Reviewer #1: **Yes:** Faraz MansoorFaraz Mansoor

Reviewer #2: No

To ensure your figures meet our technical requirements, please review our figure guidelines: http s://journals.plos.org/plosone/s/figures

You may also use PLOS’s free figure tool, NAAS, to help you prepare publication quality figures: http s://journals.plos.org/plosone/s/figures#loc-tools-for-figure-preparation.

---

## [Author Response · Author response to Decision Letter 2]

9 Mar 2026

We greatly appreciate the constructive comments and thoughtful recommendations from the academic editor and reviewers. We have addressed all comments carefully and revised the manuscript accordingly.

Response to the Academic Editor

We thank the academic editor for the careful evaluation and valuable suggestions. In addition to the translated protocol uploaded as Supplemental File S2, the original Chinese-language IRB-approved study protocol has been uploaded. On line 418, "parents" has been changed to "patients"(See page 20 line 418)

Response to Reviewer #1:

We would like to thank Reviewer #1 for the thoughtful comments and positive assessment of the revised manuscript. We appreciate the recognition of the improvements in clarity, methodological rigor, hierarchical outcomes, expanded statistical analysis plans, and enhanced data management procedures. These changes strengthen the protocol and improve transparency. As suggested, we have updated the patient recruitment date to April 2026 throughout the manuscript. (See page 6 line 115)

Response to Reviewer #2:

We appreciate the positive feedback from Reviewer #2. We are pleased that all of the comments have been addressed and incorporated into the revised manuscript.

We hope the revised manuscript meets the publication criteria of PLOS ONE.

Thank you again for your time and valuable input.

---

## [Editor Report · Decision Letter 2]

1 Apr 2026

Effects of total intravenous anesthesia on postoperative quality of recovery and the levels of inflammatory factors in patients undergoing retroperitoneal endoscopic surgery in urology: Study protocol for a randomized, controlled trial

PONE-D-25-62547R2

Dear Dr. li,

We’re pleased to inform you that your manuscript has been judged scientifically suitable for publication and will be formally accepted for publication once it meets all outstanding technical requirements.

An invoice will be generated when your article is formally accepted. Please note, if your institution has a publishing partnership with PLOS and your article meets the relevant criteria, all or part of your publication costs will be covered. Please make sure your user information is up-to-date by logging into Editorial Manager at Editorial Manager® and clicking the ‘Update My Information' link at the top of the page. For questions related to billing, please contact  and clicking the ‘Update My Information' link at the top of the page. For questions related to billing, please contact billing support..

Kind regards,

James Mockridge

Staff Editor

PLOS One
---

## [Editor Report · Acceptance letter]

PONE-D-25-62547R2

PLOS One

Dear Dr. Li,

I'm pleased to inform you that your manuscript has been deemed suitable for publication in PLOS One. Congratulations! Your manuscript is now being handed over to our production team.

Kind regards,

on behalf of

Dr James Mockridge

Staff Editor

PLOS One